# Dataflow Optimization through Exploring Single-Layer and Inter-Layer Data Reuse in Memory-Constrained Accelerators

**Jinghao Ye [1], Masao Yanagisawa [2] and Youhua Shi [2,*]**

[1]  NVIDIA Semiconductor Technology (Shanghai) Co., Ltd., Shanghai 201210, China; seiko.yo@islab.cs.waseda.ac.jp
[2]  Faculty of Science and Engineering, Waseda University, Tokyo 169-8555, Japan; myanagi@waseda.jp
[*]  Correspondence: shi@waseda.jp

**Abstract:** Off-chip memory access has become the performance and energy bottleneck in memory-constrained neural network accelerators. To provide a solution for the energy efficient processing of various neural network models, this paper proposes a dataflow optimization method for modern neural networks by exploring the opportunity of single-layer and inter-layer data reuse to minimize the amount of off-chip memory access in memory-constrained accelerators. A mathematical analysis of three inter-layer data reuse methods is first presented. Then, a comprehensive exploration to determine the optimal data reuse strategy from single-layer and inter-layer data reuse approaches is proposed. The result shows that when compared to the existing single-layer-based exploration method, SmartShuttle, the proposed approach can achieve up to 20.5% and 32.5% of off-chip memory access reduction for ResNeXt-50 and DenseNet-121, respectively.

**Keywords:** DNN accelerator; dataflow; off-chip memory access; layer fusion; data reuse





## 1. Introduction

Deep neural networks (DNNs) have been widely used in modern artificial intelligence tasks such as image recognition and segmentation. The accuracy improvement in these tasks that is achieved by DNN models such as AlexNet [1], GoogLeNet [2], ResNet [3], ResNeXt [4], and DenseNet [5] usually comes at the cost of extremely high computational complexity. These widely used DNN algorithms typically have tens of layers with tens to hundreds of megabytes (MBs) of parameters and require up to several billions of multiply-and-accumulate (MAC) computations even for single inference task, which demands both a large amount of computational hardware resources and considerable storage elements. Unfortunately, on-chip memory and the available computational resources are very limited in mobile and wearable devices (it was reported in [6] that the capacity of SRAM is typically less than 1 MB), which makes it generally impossible to save the parameters and/or the intermediate results on-chip, even for one layer. It has also been pointed out in [7] that the data movement is orders of magnitude more energy-consuming than the corresponding MAC computation. More specifically, the relative energy consumption of a 32-bit DRAM access in a 45 nm CMOS process is $6400\times$, $711\times$, $206\times$, $173\times$, and $128\times$ greater than that of a 32-bit int ADD, a 32-bit float ADD, a 32-bit int MULT, a 32-bit float MULT, and a 32-bit SRAM read operation, respectively, as shown in [8]. In addition, it has also been reported that the energy consumption of DRAM access can reach up to more than 80% of the total energy consumption in the well-known DNN accelerators, such as DianNao [9] and Cambricon-X [10]. Therefore, off-chip memory access has become the performance and energy bottleneck in DNN accelerators [11,12], and how to maximize the utilization of the already on-chip data is critical for memory-constrained accelerators.

In the literature, various techniques, such as pruning, compression, data reuse methods, etc., have been developed to reduce off-chip memory accesses for energy-efficient DNN processing. Among them, one of the most promising approaches is to leverage on-chip data

reusability, such as with input feature map reuse (***ir***) [11], partial sum reuse (***pr***) [13–15], and weight reuse (***wr***) [16,17]. These approaches have shown their advantages; however, they all consider each layer separately. Thus, no matter how large the on-chip memory is, the output feature map (ofmap) of each layer should be written to the off-chip memory and then read back as the input feature map (ifmap) of the next layer. As networks grow deeper, the amount of this shuttling data increases, leading to significant energy consumption. On the other hand, layer fusion [12] was proposed to maximize the feature map (fmap) reuse in consecutive convolutional layers, and its effectiveness on MobileNet is shown in [18]. The off-chip memory access of the weights, however, might increase. Moreover, it typically requires large on-chip memory for the storage of cross-layer feature maps, which makes it difficult to deploy it in memory-constrained designs. SuperSlash [19] adopts the exploration approach of SmartShuttle [11] and takes the advantage of layer fusion for off-chip DRAM access reduction. However, it only supports ***pr*** in the first layer and ***ir*** in the second layer of two fused layers. Moreover, it cannot support grouped convolutional layers. Consequently, the effectiveness of SuperSlash is limited in state-of-the-art neural networks, such as GoogLeNet and ResNeXt.

Although the existing works can effectively reduce the amount of off-chip memory accesses, as neural network models become more diverse for various applications, how to maintain high energy efficiency with limited hardware resources for diverse NN models is still an emerging challenge. To the best of our knowledge, no systematic approach to exploring the opportunity of both single-layer and inter-layer data reuse for minimizing off-chip memory access has yet been studied. Therefore, this paper proposes three inter-layer data reuse methods and a dataflow optimization approach to minimizing the amount of off-chip memory access in memory-constrained accelerators, with the following contributions:

(1) A mathematical analysis in terms of required off-chip memory access and on-chip memory capacity for the proposed inter-layer data reuse methods is introduced for modern neural networks. Unlike most of the existing data reuse methods, in which only AlexNet and a VGG-like structure are considered, our analysis can be applied to most of the existing convolutional neural networks, ranging from AlexNet, VGG, ResNet, and DenseNet to ResNeXt, etc.

(2) A comprehensive exploration to precisely determine the optimal data reuse strategy from various single-layer and inter-layer-based reuse approaches is proposed for memory-constrained DNN accelerators. Unlike the single-layer-based SmartShuttle and the layer-fusion-based SuperSlash, our method can determine the optimal data flow and the corresponding data reuse strategy along layers; thus, the proposed method can always achieve minimum DRAM access for accelerators with various SRAM capacity.

The rest of this paper is organized as follows. Section 2 gives the background. The mathematical analysis of the inter-layer data reuse methods is introduced in Section 3, and the optimal dataflow exploration method is illustrated in Section 4. The evaluation results are presented in Section 5. Finally, Section 6 concludes this paper.

## 2. Preliminaries

### 2.1. DNN Accelerators

DNN accelerators have been developed with various design approaches [7–26]. Due to the data-centric property in recent ASIC-based DNN accelerators, in which a significantly large amount of data should be processed and transferred in and out of the accelerator chips, memory plays an important role. The typical on-chip global memory architectures can be simply classified into two types, i.e., those which use a unified buffer, such as those in [13,16,26], and those which use separate buffers for input feature maps, filter weights, and partial sums, such as those in [15,17]. Using a multi-bank-based unified global buffer can flexibly change the volume of the on-chip ifmaps, weights, and psums in different layers, while using separated buffers can transact different types of data in parallel.

Recently, a layer-fusion-based DNN accelerator was presented in [18]; it stores the relevant data of two consecutive layers to support 2-layer fusion, and its architecture is similar to the unified global buffer-based architecture. Therefore, considering the amount of off-chip memory access among the layers (i.e., there are typically more fmaps in the shallow layers while there are more weights in the deeper layers), we adopt the architecture with a unified buffer for optimal dataflow exploration in memory-constrained accelerators.

### 2.2. Single-Layer-Based Data Reuse

The technique called dataflow has been widely explored for efficiency improvement in DNN accelerator designs; it not only includes how to partition a large amount of off-chip data into small tiles to fit on-chip memory, but also determines how data move in the memory hierarchy. Off-chip memory access causes significant energy consumption; therefore, most existing dataflows focus on increasing data reuse efficiency for off-chip memory access reduction.

Three kinds of data reuse strategies, such as *ir*, *wr*, and *pr*, have been proposed with the basic idea of maximizing the utilization efficiency of the on-chip data if they are fetched from off-chip memory. For example, in *ir*, once an ifmap tile is read, it stays on-chip and will not be discarded until all the computations related to it are completed. By doing this, we can maximize the utilization of the loaded ifmap; thus, each ifmap datum only needs to be fetched once. Similarly, the weights are only read once in *wr*, while *pr* focuses on eliminating the movement of psums. It should be noted that most of the existing methods conduct data reuse in a layer-based manner, while data reusability in consecutive layers has not been considered. Therefore, they can be viewed as single-layer-based data reuse. The tiling of single-layer-based convolution is shown in Figure 1, with the corresponding description of the network-defined shape parameters and tiling parameters given in Tables 1 and 2, respectively.

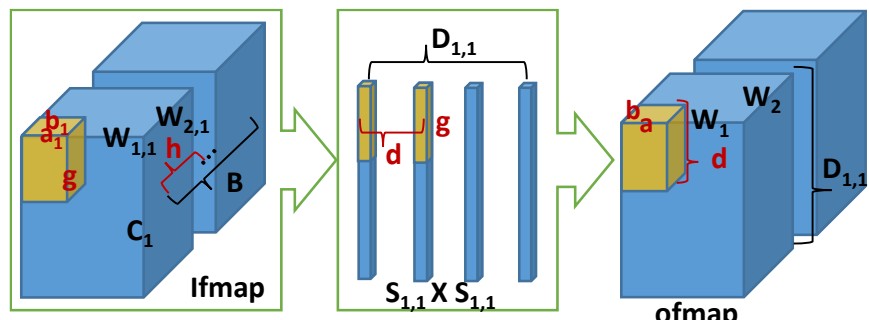

**Figure 1.** Tiling of single-layer-based convolution.

**Table 1.** Shape parameters of conv layers.

| Shape Parameter | Description |
| --- | --- |
| $W_{1,i}/W_{2,i}$ | Ifmap width/height in the $i$th layer |
| $W_1/W_2$ | Ofmap width/height of last layer |
| $B$ | Batch size |
| $C_1$ | Input channels |
| $C$ | Dimension of Sublayers (Cardinality) |
| $D_{i,j}$ | Filters of the $i$th layer in the $j$th sublayer |
| $S_{i,j} \times S_{i,j}$ | Filter size of the $i$th layer in the $j$th sublayer |
| $Pool_i$ | Pooling stride of $i$th layer |
| $P_i$ | Stride of the $i$th layer |

**Table 2.** Tiling parameters of dataflow.

| Tiling Parameter | Description |
|:---:|:---:|
| $a,b$ | Weight/height of on-chip ofmap |
| $h$ | On-chip batch size |
| $g$ | On-chip ifmap channels |
| $c$ | On-chip sublayers |
| $d$ | On-chip filters in the first layer |
| $e$ | On-chip filters in the second layer |

To move huge amounts of data from a large memory (i.e., off-chip DRAM) into a small one (i.e., an on-chip global buffer), we typically partition the data into very small tiles. According to the tiling parameters shown in Table 2, without any data reuse, we need to read each ifmap tile $\lceil D/d \rceil$ times; each weight tile $\lceil W_1 W_2 B / abh \rceil$ times; and write each ofmap once in a single-layer-based dataflow. Because psums should be written to the DRAM and then read back for accumulation, the required amount of DRAM access (DA) for each ofmap is $2(\lceil C_1 / g \rceil - 1)$. Therefore, the tiling parameters can be determined according to the desired data reuse strategy. For example, if the tiling parameter ($d$) is chosen to be D (i.e., the corresponding part of all the filters are fetched to on-chip memory), the off-chip memory access of each ifmap can be reduced to be 1, which indicates that the ifmap reuse efficiency is maximized.

To determine the optimal data reuse strategy with proper tiling parameters, SmartShuttle, a layer-wise data reuse exploration method, was proposed in [11]. Unlike previous approaches [10,13–15], in which all the layers adopt the same data reuse strategy, SmartShuttle can adaptively select the most suitable data reuse strategy for each layer. In addition to SmartShuttle, the recent works [24,25,27] also belong to adaptive layer-wise data reuse approaches. They, however, still fall into the category of single-layer-based data reuse, in which no matter how large the on-chip memory is, the generated ofmap of the current layer should be stored to off-chip memory and then read back as the input of the next layer. As the networks grow deeper, the amount of this shuttling data increases, leading to larger energy consumption.

Layer fusion was proposed in [12] to maximize the feature map reuse in consecutive layers, which is suitable for modern networks with networks-in-network and $1 \times 1$ convolutions. However, a large amount of on-chip memory is generally required; for instance, the accelerator presented in [18] has 2 MB on-chip memory for MobileNet V1 with 2-layer fusion. An adaptive weight reuse method for shortcut layer data was proposed in [28], following from SmartShuttle, but was trying to solve the problem in SmartShuttle in which the amount of DRAM access cannot be further reduced even with large on-chip memory. As a result, as with layer fusion [12], significantly large on-chip memory (in several MBs) is required, which limits the utilization of these inter-layer reuse-based methods in low-cost memory-constrained designs. Moreover, few works have been presented with precise mathematical analysis to obtain the optimal dataflow for layer-fusion-based data reuse, which motivates us to explore the opportunity of inter-layer data reuse for minimizing off-chip memory access.

## 3. Inter-Layer Data Reuse

Unlike with single-layer-based data reuse approaches, to improve the efficiency of data reuse across layers we should carefully study the data dependency in the basic module of each network model. Some typical modules in modern networks, such as the normal CONV layers in AlexNet and VGG, the Inception modules in GoogLeNet, and the Bottleneck blocks in ResNet and Network-in-Neuron in ResNext, are shown in Figure 2, in which, if two or more consecutive layers can be fused together, the corresponding off-chip memory access of the inter-layer feature maps can be eliminated. Moreover, if the input feature maps of the grouped convolutional layers, such as the Inception module and Network-in-Neuron shown in Figure 2c,d, can be reused, the off-chip memory access can also be reduced. Thus,

without loss of generality, grouped convolutional layers, such as those in the Inception module and Network-in-Neuron, are used as the basic module for exploration in our work as it can be easily transformed to other ones, such as the normal CONV layers in VGG and the Bottleneck block in ResNet.

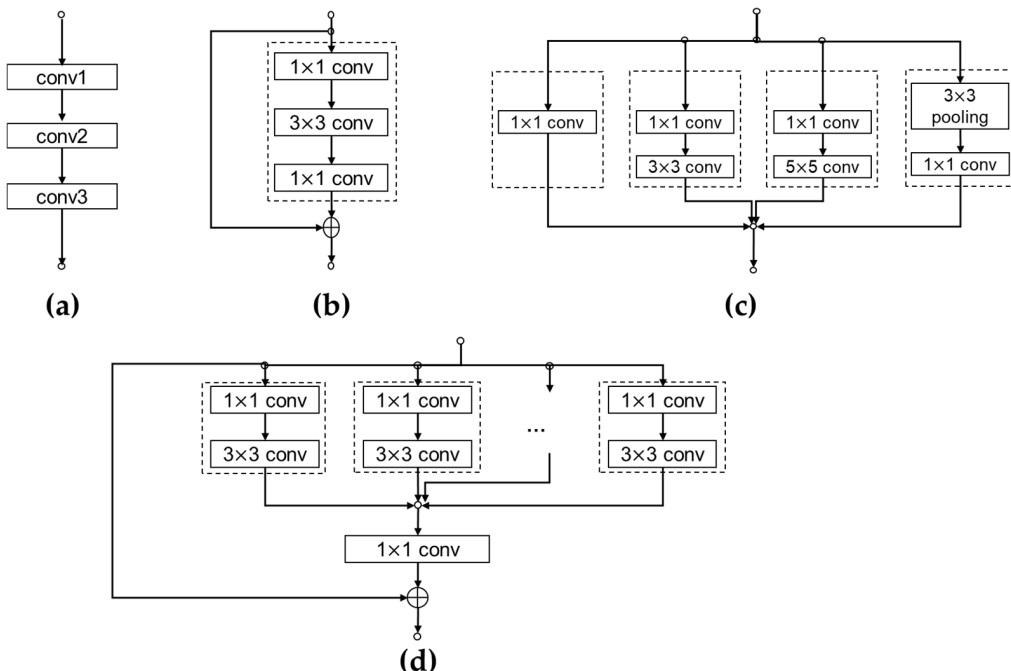

**Figure 2.** Typical modules in modern networks: (**a**) normal CONV layers such as those in AlexNet and VGG; (**b**) Bottleneck block in ResNet; (**c**) Inception module in GoogLeNet; and (**d**) Network-in-Neuron in ResNeXt.

According to the discussion on single-layer-based dataflow, the optimal dataflow problem can be thought of as how to partition ifmaps and weights into small tiles for efficient data reuse so as to minimize the total amount of off-chip memory access with the limited capacity of on-chip memory. As with the single-layer-based dataflow shown in Figure 1, the dataflow of the fused 2-layer (F2L) is shown in Figure 3, in which there are $C$ sublayers (i.e., Cardinality), and each of them has two layers, indicated as Layer 1 and Layer 2. It is worth noting that although this paper mainly discusses data reuse in two fused CONV layers, the idea of the proposed analysis and exploration can be extended to the cases with more CONV layers being fused together in a similar way and can also be applied to be combined with CONV layers with FC layers and pooling layers.

As shown in Figure 3, in order to generate the $W_1 \times W_2$ ofmap values of Layer 2, with the consideration of zero-padding, the required ifmaps of Layer 2 (i.e., the ofmap of Layer 1) need to be $(P_2 W_1 \cdot Pool_2 + max(S_{2,x}) - P_2) \times (P_2 W_2 \cdot Pool_2 + max(S_{2,x}) - P_2)$, which is represented as $W_{1,2} \times W_{2,2}$ in the figure, and consequently, the required ifmaps of Layer 1 denoted as $W_{1,1} \times W_{2,1}$ should be $(P_1 W_{1,1} \cdot Pool_1 + max(S_{1,x}) - P_1) \times (P_1 W_{2,1} \cdot Pool_1 + max(S_{1,x}) - P_1)$. If these ifmaps can be fetched in an optimal way, the corresponding off-chip memory access of the inter-layer ofmap/ifmap can be eliminated in F2L. Moreover, because all the sublayers have the same ifmap, the ifmap movement can be reduced through exploring the parallelism of the sublayers under the constraint of the on-chip memory capacity. Because the off-chip memory access of the ifmap data of Layer 1 depends on the ratio of the total amount of weights of the layer to the size of the weight tiles, without proper data reuse, each ifmap and each weight need to be read from the off-chip memory $\lceil CD_1/cd \rceil$ and $\lceil W_1 W_2 B/abh \rceil$ times, respectively.

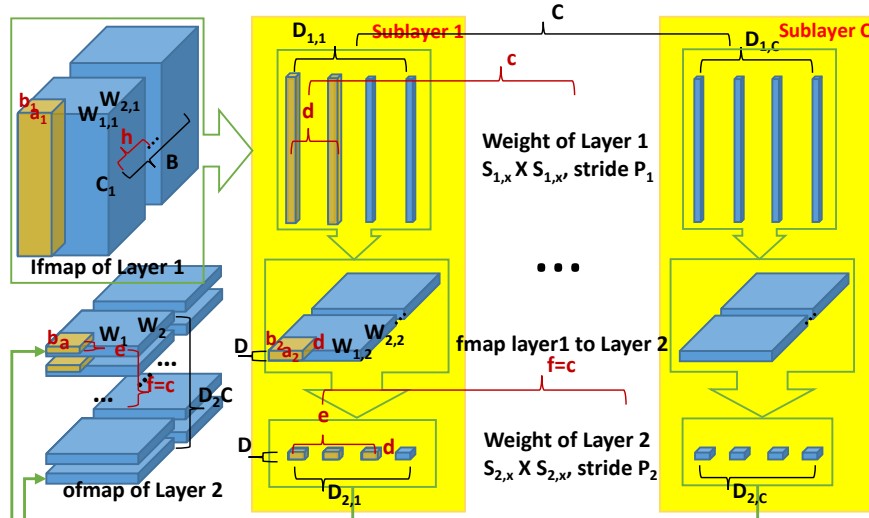

**Figure 3.** Tiling and dataflow in inter-layer reuse.

Unlike the existing layer-fusion-based methods [12,18], in which only fmap reuse is considered, the proposed strategies for the various data reuse in the fused 2-layer are illustrated in the following. Here, we assume that the ifmap tile of Layer 1 should cover all the $C_1$ channels, as shown in Figure 3; thus, the size of an ifmap tile can be expressed as $a_1 b_1 C_1$.

- **Ifmap reuse in fused two layers**

We first introduce the accurate mathematical analysis for ifmap reuse in the fused 2-layer (*ir2l*). The strategy of *ir2l* is to store as many ifmaps of Layer 1 as possible and to ensure that the inter-layer feature maps and all the generated psums can be stored in a global buffer, while the rest of the global buffer can be used to store as many weights as possible to increase the degree of parallelism and speed up the operation.

As with the existing single-layer-based data reuse schemes, *ir2l* also has four stages: (1) the ifmaps and weights of Layer 1 are loaded from the off-chip memory (i.e., DRAM) to an on-chip global buffer; (2) the ifmaps and weights of Layer 1 are transferred to a local buffer, and the convolutional computations are conducted by fully reusing the on-chip ifmaps to generate the ofmap values of Layer 1 (i.e., the ifmap values of Layer 2); (3) the ofmap of Layer 1 is saved on-chip, and the required weights of Layer 2 for the generated ofmaps of Layer 1 are read; and (4) the ofmap values of Layer 2 are generated and written to the off-chip DRAM.

The corresponding pseudo-code of *ir2l* is shown in Figure 4a, where $NT_{ifmap} = W_1 W_2 B / abh$. The pseudo-code of *ir2l* contains four loops, and the outermost loop shows how the ifmaps are reused in two fused layers; this indicates that each ifmap of Layer 1 needs to be loaded only once, and the ofmaps of Layer 1 (i.e., ifmaps of Layer 2) are kept on-chip for inter-layer feature map reuse.

In *ir2l*, an ifmap tile of Layer 1 with the size of $a_1 b_1 C_1 h$ is first read from the DRAM. After transmitting the ifmap tile into the on-chip global buffer, one filter of the *j*th sublayer in Layer 1, with the size of $S_{1,x} S_{1,x} C_1$ where $x = j$, is fetched. With these ifmaps and filters, the corresponding ofmap data of Layer 1 (i.e., the ifmap of Layer 2 with the size of $a_2 b_2 h$) can be generated. Then, the corresponding filers of Layer 2 will be read from the DRAM. Because only one channel of the imaps of Layer 2 is on-chip, only one channel in the filter of the *j*th sublayer in Layer 2 (in size of $S_{2,x} \times S_{2,x}$) is required to be fetched for the convolutional computations. Consequently, by using the on-chip ifmaps and weights, the corresponding psum of Layer 2 with the size of $abh$ can be generated and saved on-chip. Then, by reading only one new channel of the next filter of Layer 2 from the DRAM, another psum of Layer 2 can be generated by using the already on-chip ifmaps. Finally, the $abhD_{2,j}$

psums of Layer 2 can be generated in this loop, and the corresponding on-chip buffer should be reserved for the storage of these psums for further accumulation. When the on-chip ifmaps of Layer 2 have been completely reused, they are discarded, and new filters of Layer 1 are fetched for the generation of the new ofmaps of Layer 1. After the corresponding $abhD_{2,j}$ psums of Layer 2 have been fully accumulated, the results are transmitted to the DRAM as the ofmap tile in the $j$th sublayer. When the on-chip ifmaps of Layer 1 have been fully reused, the next ifmap tile is loaded and the above steps are repeated.

```
for ( m = 0 , m < NT_ifmap , m ++ ) {
    read ifmaps;
    for ( j = 0 , j < C , j ++ ) {
        for ( i = 0 , i < D_1,j , i ++ ) {
            read weights of Layer 1;
            out_layer1[a_2,b_2,h] = ifmap[a_1,b_1,C_1,h] * filter_layer1[S_1,j,S_1,j,C_1] ;
            for ( k = 0 , k < D_2,j , k ++ ) {
                read weights of Layer 2;
                psum_layer_2[a,b,h,k] = out_layer1[a_2,b_2,h] * filter_layer2[S_2,j,S_2,j] ;
            }
            accumulate psums of Layer 2 in local psum buffers;
        }
        write ofmap of Layer 2 to DRAM;
    }
}
```

(a)

```
for ( l = 0 , l < C/c , l ++ ) {
    read weights of Layer 1;
    read weights of Layer 2;
    for ( m = 0 , m < NT_ifmap , m ++ ) {
        read ifmaps;
        for ( j = 0 , j < c , j ++ ) {
            for ( i = 0 , i < D_1,j , i ++ ) {
                out_layer1[a_2,b_2,h] = ifmap[a_1,b_1,C_1,h] * filter_layer1[S_1,j,S_1,j,C_1] ;
                for ( k = 0 , k < D_2,j , k ++ ) {
                    psum_layer_2[a,b,h,k] = out_layer1[a_2,b_2,h] * filter_layer2[S_2,j,S_2,j] ;
                }
                accumulate psums of Layer 2 in local psum buffers
            }
            write ofmap of Layer 2 to DRAM
        }
    }
}
```

(b)

```
for ( j = 0 , j < C , j ++ ) {
    for ( l = 0, l < D_1,j , l ++ ) {
        read weights of Layer 1;
        read weights of Layer 2;
        for ( m = 0 , m < NT_ifmap , j ++ ) {
            read ifmaps from DRAM into global buffer;
            for ( i = 0 , i < d , i ++ )
                out_layer1[a_2,b_2,h] = ifmap[a_1,b_1,C_1,h] * filter_layer1[S_1,j,S_1,j,C_1] ;
                for ( k = 0 , k < D_2,j , k ++ ) {
                    psum_layer_2[a,b,h,k] = out_layer1[a_2,b_2,h] * filter_layer2[S_2,j,S_2,j] ;
                }
                accumulate psums of Layer 2 in local psum buffers;
        }
        write psums of Layer 2 to global buffer;
    }
}
write ofmap of Layer 2 to DRAM
```

(c)

```
for ( j = 0 , j < C , j ++ ) {
    for ( m = 0 , m < NT_ifmap , j ++ ) {
        read ifmaps from DRAM;
        for ( l = 0 , l < D_1,j , l ++ )
            read weights of Layer 1 from DRAM;
            read weights of Layer 2 from DRAM;
            for ( i = 0 , i < d , i ++ )
                out_layer1[a_2,b_2,h] = ifmap[a_1,b_2,C_1,h] * filter_layer1[S_1,j,S_1,j,C_1] ;
                for ( k = 0 , k < D_2,j , k ++ ) {
                    psum_layer_2[a,b,h,k] = out_layer1[a_2,b_2,h] * filter_layer2[S_2,j,S_2,j] ;
                }
                accumulate psums of Layer 2 in local psum buffer;
            write psums of Layer 2 to global buffer;
        }
        write ofmap of Layer 2 to DRAM;
    }
}
```

(d)

**Figure 4.** Pseudo-code of various data reuse strategies in fused 2-layers: (**a**) *ir2l*, (**b**) *wr2lv1*, (**c**) *wr2lv2*, and (**d**) *pr2l*.

For efficient ifmap reuse in the fused layers, it is desired that both the required ifmap data of Layer 1 and the corresponding psum data of Layer2 should be kept on-chip. Let the ifmap tile of Layer 1 be in the size of $a_1 b_1 C_1$ and thus with the consideration of padding $a_1 \times b_1 = (P_1(P_2 a \cdot Pool_2 + max(S_{2,x}) - P_2) \cdot Pool_1 + max(S_{1,x}) - P_1) \times (P_1(P_2 b \cdot Pool_2 + max(S_{2,x}) - P_2) \cdot Pool_1 + max(S_{1,x}) - P_1)$. It is worth noting that here $max(S_{1,x})$ and $max(S_{2,x})$ indicate the largest filters in Layer 1 and 2, respectively, which is because the filter size in grouped convolution might be different, as with that in GoogLeNet, as shown in Figure 2c. As the transaction frequency of the weights depends on the ratio of the on-chip ifmap data to the total ifmap data of Layer 1, $c$, $d$, and $e$ can even be set as 1 for the saving of the required on-chip resources for the storage of ifmaps and psums.

Assuming that the number of output channels of a sublayer is $D_{2,x}$ and that $max(D_{2,x})$ indicates the maximum output channels among all the sublayers, the global buffer must be large enough to store $abh \cdot max(D_{2,x})$ psums. In addition, it is necessary to store $a_1 b_1 C_1 h$ ifmaps, $a_2 b_2 h$ inter-layer feature maps (fmaps) from Layer 1 to Layer 2, where $a_2 b_2 = (Pa + max(S_{2,x}) - P)(Pb + max(S_{2,x}) - P)$ with the consideration of padding, and

the maximum weights of a layer in specific channels among all the sublayers. Therefore, the total amount of off-chip DRAM access in *ir2l* ($DA_{ir2l}$) for all the ifmaps/weights/ofmaps is:

$$DA_{ir2l} = \frac{BW_1W_2}{abh}\left(C_1\sum_{x=1}^{C}D_{1,x}S_{1,x}^2 + \sum_{x=1}^{C}D_{1,x}D_{2,x}S_{2,x}^2\right) + \frac{BW_1W_2C_1}{ab}a_1b_1 + \sum_{x=1}^{C}D_{2,x}W_1W_2B \tag{1}$$

and the required on-chip global buffer capacity is:

$$S_{cir2l} = a_1b_1C_1h + a_2b_2h + abhD_{2,x} + max(C_1S_{1,x}^2, S_{2,x}^2) \tag{2}$$

- **Weight reuse in fused two layers**

Similarly, weight reuse in the fused 2-layer (*wr2l*) is also possible. However, the capacity of the on-chip memory will affect the dataflow, depending on whether the weights of more than one sublayer can be stored on-chip or not. To store all the weights of more than one sublayer requires a large amount of global buffer capacity, but if it is possible, the ofmap values of Layer 2 can be generated by using only the on-chip data, which can help to save the amount of global buffer used for psum storage. On the other hand, if only a part of the weights of one sublayer are stored on-chip, more on-chip memory is required to store the intermediate psums. As networks become more diverse, we cannot expect to obtain satisfying results by storing all the required weights on-chip, especially for the cases in the fused layers. Therefore, the accurate analysis and formulation of off-chip memory access with given on-chip memory capacity are provided in the following.

**Case 1: Weights of more than one sublayer can be stored on-chip (*wr2lv1*)**

The points of this data reuse strategy fall into two folds: to store as many weights as possible in the global buffer and to save as many ifmaps of Layer 1 as possible on-chip. In this case, if the weights of at least one sublayer are completely stored on-chip, the ofmaps can be generated by only using the on-chip data. Although the storage of these weights is required, the total amount of on-chip memory used for the psum storage might be reduced.

The pseudo-code of *wr2lv1* is shown in Figure 4b. There are five loops in *wr2lv1*, and the outermost loop shows how the weights of the two fused layers are reused, in which each weight only needs to be fetched once. Initially, the $C_1\sum_{x=1}^{c}D_{1,x}S_{1,x}^2$ weights of Layer 1 and the $\sum_{x=1}^{c}D_{1,x}D_{2,x}S_{2,x}^2$ weights of Layer 2 are fetched from the off-chip memory. After that, the $a_1b_1C_1h$ ifmaps of Layer 1 are also fetched. Because the frequency of the data transaction to read in the ifmaps depends on the on-chip weights and each weight should be read from the off-chip memory only once in weight-reuse-based dataflow, the parameter $h$ can therefore be set as small as 1. However, if $a$ and $b$ are too small, $W_1W_2a_1b_1/ab$ might be greater than $W_1W_2$, which slightly influences the off-chip memory access, and the saved on-chip memory can be used to store more weights for further DRAM access reduction.

In general, the global buffer needs to store $C_1\sum_{x=1}^{c}D_{1,x}S_{1,x}^2$ weights of Layer 1 and $\sum_{x=1}^{c}D_{1,x}D_{2,x}S_{2,x}^2$ weights of Layer 2; $a_1b_1C_1h$ ifmaps of Layer 1; and the $a_2b_2h$ interlayer fmaps. Therefore, the required amount of off-chip memory access of *wr2lv1* can be expressed as

$$DA_{wr2lv1} = \left(C_1\sum_{x=1}^{C}D_{1,x}S_{1,x}^2 + \sum_{x=1}^{C}D_{1,x}D_{2,x}S_{2,x}^2\right) + \frac{BW_1W_2C_1C}{abc}a_1b_1 + \sum_{x=1}^{C}D_{2,x}W_1W_2B \tag{3}$$

and the required capacity of the on-chip global buffer is

$$S_{ciwr2lv1} = a_1b_1C_1h + a_2b_2h + \left(C_1\sum_{x=1}^{c}D_{1,x}S_{1,x}^2 + \sum_{x=1}^{c}D_{1,x}D_{2,x}S_{2,x}^2\right) \tag{4}$$

**Case 2: Weights of less than one sublayer can be stored on-chip (*wr2lv2*)**

The strategy of *wr2lv2* is to store as many weights in the global buffer as possible, while keeping enough storage space for the intermediate generated psums of Layer 2 and storing as many ifmaps of Layer 1 as possible. In this case, because the on-chip global

buffer only stores weights of less than one sublayer, it needs to keep sufficiently large global buffer resources to store the corresponding psums to avoid off-chip memory accesses of the psums. The pseudo-code of *wr2lv2* is shown in Figure 4c.

In this case, the global buffer needs to store $C_1 d S_{1,x}^2$ weights for Layer 1; $dD_{2,x}S_{2,x}^2$ weights for Layer 2; the $a_1 b_1 C_1 h$ ifmaps; the $a_2 b_2 h$ inter-layer fmaps; the and $BW_1 W_2 D_{2,x}$ psums; therefore, the amount of DRAM accesses and the capacity of the global buffer of *wr2lv2* are:

$$DA_{wr2lv2} = \left( C_1 \sum_{i=1}^{C} D_{1,x} S_{1,x}^2 + \sum_{i=1}^{C} D_{1,x} D_{2,x} S_{2,x}^2 \right) + \frac{BW_1 W_2 C_1}{abd} \sum_{i=1}^{C} D_{1,x} a_1 b_1 + \sum_{x=1}^{C} D_{2,x} W_1 W_2 B \tag{5}$$

and

$$S_{cwr2lv2} = a_1 b_1 C_1 h + a_2 b_2 h + BW_1 W_2 D_{2,x} + \left( C_1 d S_{1,x}^2 + d D_{2,x} S_{2,x}^2 \right) \tag{6}$$

- **Psum reuse in fused two layers**

The psum reuse in fused 2-layer (*pr2l*) is to maximize the reuse of psums. Storing the relevant data of the integral sublayers can directly generate ofmaps of Layer 2, which makes it possible not to store additional psums in a global buffer. Therefore, psum reuse is only applied for the case in which weights of more than one sublayer can be stored on-chip. The corresponding pseudo-code of *pr2l* is shown in Figure 4d.

In *pr2l*, the $C_1 d S_{1,x}^2$ weights of Layer 1 and the $dD_{2,x} S_{2,x}^2$ weights of Layer 2 should be stored in a global buffer; they are read from the off-chip memory $W_1 W_2 B/abh$ times. The amount of on-chip ifmaps is $a_1 b_1 C_1 h$, and the same ifmaps of Layer 1 need to be read from the off-chip memory $C$ times. Therefore, in *pr2l*, no psums need to communicate with the off-chip memory, while both the weights and the ifmaps need to be read several times. Thus, the tiling size should be carefully selected for off-chip memory access minimization. Generally, the global buffer needs to store $C_1 d S_{1,x}^2$ weights for Layer 1; $dD_{2,x} S_{2,x}^2$ weights for Layer 2; $a_1 b_1 C_1 h$ ifmaps of Layer 1; $a_2 b_2 h$ inter-layer fmaps; and $abhD_{2,x}$ psums. Thus, the required DRAM access ($DA_{pr2l}$) and on-chip memory capacity ($S_{cpr2l}$) are:

$$DA_{pr2l} = \frac{BW_1 W_2}{abh} \left( C_1 \sum_{x=1}^{C} D_{1,x} S_{1,x}^2 + \sum_{x=1}^{C} D_{1,x} D_{2,x} S_{2,x}^2 \right) + \frac{BW_1 W_2 C_1 C}{ab} a_1 b_1 + \sum_{x=1}^{C} D_{2,x} W_1 W_2 B \tag{7}$$

and

$$S_{cpr2l} = a_1 b_1 C_1 h + a_2 b_2 h + abhD_{2,x} + \left( C_1 d S_{1,x}^2 + d D_{2,x} S_{2,x}^2 \right) \tag{8}$$

## 4. Optimal Dataflow Exploration for Hybrid Data Reuse

Up till now, the comprehensive mathematical formulation for the inter-layer data reuse approaches have been derived, which makes it possible for us to conduct a design space exploration to determine the optimal inter-layer data reuse strategy with the corresponding tiling parameters for minimizing off-chip memory access while meeting the capacity constraint of on-chip memory. The optimization problem for inter-layer data reuse is formulated as follows:

$$DA_{f2l} = \min \left( DA_{ir2l}, DA_{wr2lv1}, DA_{wr2lv2}, DA_{pr2l} \right) \text{subject to} \begin{cases} S_{c_{ir2l}} \leq SRAM_c \\ S_{c_{wr2lv1}} \leq SRAM_c \\ S_{c_{wr2lv2}} \leq SRAM_c \\ S_{c_{pr2l}} \leq SRAM_c \end{cases} \tag{9}$$

where $DA_{f2l}$ and $SRAM_c$ represent the minimum off-chip memory access in F2L and the maximum capacity of the global buffer in a specified DNN accelerator, respectively.

Considering the diversity of neural networks, we extend the exploration space for further optimization, and the improved formulation to determine the optimal data reuse

strategy from single-layer-based and inter-layer-based data reuse approaches is given as below, with the concept of this hybrid data reuse approach shown in Figure 5.

$$DA_{optimal} = \min\left(DA_{f2l}, DA_{2sl}\right) \text{subject to} \begin{cases} S_{c_{f2l}} \leq SRAM_c \\ S_{c_{2sl}} \leq SRAM_c \end{cases} \quad (10)$$

where $DA_{f2l}$ and $DA_{2sl}$ represent the minimum off-chip memory access required in inter-layer data reuse (i.e., fused 2-layers) and that using the single-layer-based data reuse, respectively. In the exploration of single-layer-based data reuse approaches, *ir*, *wr*, and *pr* are explored layer-by-layer; thus, we can adaptively determine the optimal data reuse strategy for each layer. According to the above formulas, we can determine the optimal data reuse strategy with the corresponding tiling parameters $\langle a, b, c, d, e, g, h \rangle$ from single-layer-based and inter-layer-based data reuse approaches for DNN accelerators under the constraint of on-chip storage capacity.

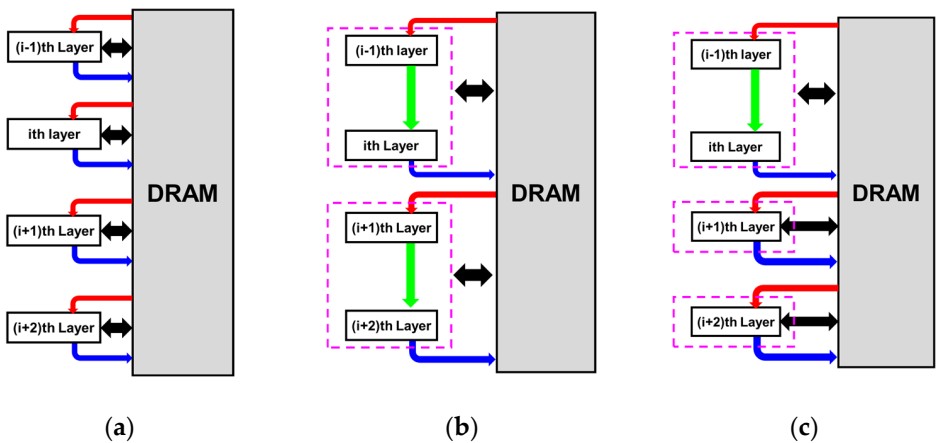

(a)  (b)  (c)

**Figure 5.** Concepts of various dataflows: (**a**) single-layer data reuse, (**b**) layer fusion in [18], and (**c**) proposed hybrid reuse.

In our work, the optimal data reuse strategy with the corresponding tiling parameters are obtained through the optimal dataflow exploration with a brute force search. In the single-layer-based exploration, in order to minimize the total amount of off-chip memory access, all the possible data reuse strategies with the corresponding tiling sizes will be explored. Because the exploration is conducted with the given constraints, the design space is not large, and the exploration can be finished in seconds. On the other hand, with regard to the inter-layer-based data reuse exploration, it will become much more complex because we need to consider how many, and which, layers can be fused. To reduce the computation complexity, this work only focuses on the exploration of the optimal dataflow of two consecutive convolutional layers. Unlike single-layer-based exploration, inter-layer-based data reuse exploration is conducted in a back-and-forth way; thus, the computation complexity is reduced.

Figure 5 illustrates the concept difference between the single-layer-based dataflow, the layer-fusion approach, and the proposed optimal hybrid approach, from which it is clear that as a more flexible approach, the proposed method promises to outperform the existing methods with less off-chip memory access.

## 5. Evaluation and Comparison Results

With all the derived formulations as shown above, the proposed optimal dataflow exploration method is built in Python, which takes (i) the layer information of a target neural network and (ii) the memory constraint of a DNN accelerator as inputs. In our work, the exploration is performed in an exhaustive search manner for minimum DRAM accesses. As an output, the optimal dataflow with the corresponding tiling configurations will be generated.

To confirm the effectiveness of the proposed optimal exploration method over the state-of-the-art exploration methods, we run evaluations by using three popular modern networks, DenseNet-121 [5], ResNeXt-50 [4], and AlexNet [1]. It is worth noting that because nonlinear operations, such as ReLU, pooling, and BN, can be performed on-chip, this work only focuses on the exploration of the optimal dataflow of two consecutive convolutional layers.

- **DenseNet-121**

DenseNet was proposed by Huang et al. in [5]. The basic module in DenseNet, the DenseLayer in each Dense Block, is similar to the traditional modules, such as those in AlexNet and VGG, except that each layer in DenseNet takes all the outputs of the preceding layers as its input. Thus, the shape parameter $C$ used in DenseNet is 1. The selection of DenseNet is due to the high classification accuracy and its dense connection between layers.

As an example, Figure 6 shows the exploration results for the 8th DenseLayer in Dense Block (2) and the last DenseLayer in Dense Block (4), both of which contain $1 \times 1$ and $3 \times 3$ convolution layers, respectively, with the global buffer capacity ranging from 32 KB to 512 KB.

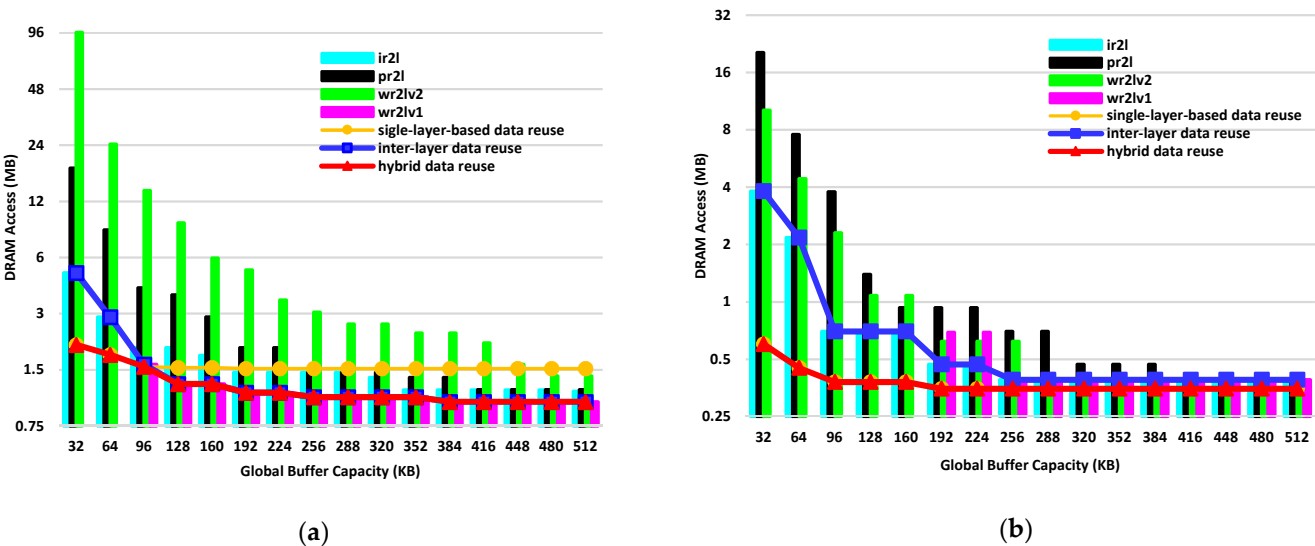

(**a**)　　　　　　　　　　　　　　　　(**b**)

**Figure 6.** DRAM access of DenseNet-121 with various global buffer capacities: (**a**) the eighth $1 \times 1$, $3 \times 3$ convolution layers in Dense Block (2) and (**b**) the last $1 \times 1$, $3 \times 3$ convolution layers in Dense Block (4).

For the results in Figure 6a, the ofmap size of Layer 2 ($W_1 \times W_2$) is $28 \times 28$; the number of input channels of Layer 1 ($C_1$) is 336; the number of output channels ($D_1$) of Layer 1 (i.e., the input channels of Layer 2) is 128; and the number of output channels ($D_2$) of Layer 2 is 32. The filter sizes of Layer 1 and Layer 2 ($S_1 \times S_1$ and $S_2 \times S_2$) are $1 \times 1$ and $3 \times 3$, respectively, and the batch size ($B$) is 3. For the results of the inter-layer data reuse methods (i.e., ***ir2l***, ***wr2lv1***, ***wr2lv2***, and ***pr2l***), when the size of global buffer is smaller than 64 KB, ***ir2l*** achieves the minimized off-chip memory access; while when the size of global buffer is greater than 96 KB, ***wr2lv2*** has the best result. This explains that the proper inter-layer data reuse approach should be adaptively selected under the constraint of the on-chip memory capacity. On the other hand, when compared to the best results of single-layer-based data reuse (i.e., ***ir***, ***pr***, and ***wr***), it is obvious that single-layer-based data reuse always has less off-chip memory access than the inter-layer data reuse methods when the on-chip memory is small, while the inter-layer data reuse methods have better results if given a large on-chip memory. Due to the variations of the off-chip memory access in modern neural networks, this confirms that we need to select the best approach from the single-layer and inter-layer-based data reuse methods for minimizing off-chip memory access.

Figure 6b shows the results of the last DenseLayer in Dense Block (4) of DenseNet-121, in which $W_1 = W_2 = 7$, and $C_1 = 988$. Unlike the results shown in Figure 6a, the optimal single-layer-based data reuse method always outperforms the inter-layer-based data reuse methods for the cases when the global buffer capacity ranges from 32 KB to 512 KB. In the case of inter-layer data reuse with two fused layers, for each ofmap value of Layer 2 the required ifmap values of Layer 1 in F2L should be $9 \times 9 \times 988$, which is because the filters in the two consecutive layers are $1 \times 1$ and $3 \times 3$, respectively. On the other hand, in the single-layer-based reuse approaches, the corresponding ifmap size is only $7 \times 7 \times 988$, as the filter size in Layer1 is $1 \times 1$. Therefore, even though the single-layer-based reuse approach needs to read and write $7 \times 7 \times 128$ fmaps from Layer 1 to Layer 2, the total amount of off-chip memory accesses is smaller than that of the inter-layer-based methods. This example illustrates that, even though the global buffer is sufficiently large, inter-layer-based reuse methods cannot always outperform the single-layer-based reuse approaches. Therefore, it is necessary to perform accurate mathematical analysis on target DNN models to determine the optimized dataflow (single-layer or inter-layer-based data reuse strategy with the corresponding tiling parameters) layer by layer for off-chip memory access minimization in designs with limited on-chip memory capacity.

Figure 7 provides the results on the total amount of off-chip memory access of DenseNet-121 with $k = 32$ and $\theta = 0.5$ [4] under the various capacity constraints of the on-chip global buffer. To gain more insights, the results of the existing single-layer-based exploration method, SmartShuttle [11], are also provided for comparisons. Consistent with the results shown in Figure 6a, inter-layer reuse with two fused layers outperforms single-layer reuse with SmartShuttle when the capacity of the on-chip memory is larger than 128 KB, while single-layer reuse is suitable for DNN accelerators with small on-chip memory. Moreover, the proposed hybrid data reuse approach outperforms single-layer reuse with SmartShuttle and inter-layer reuse with two fused layers in all the cases, especially in the range that is close to the intersection of the two curves. With a 128 KB on-chip memory, the hybrid data reuse can achieve 24.3% of off-chip memory access reduction when compared to the two methods. For the capacity of on-chip memory ranging from 64 KB to 512 KB, the hybrid data reuse approach can achieve up 32.5% and 48.7% of off-chip memory access reduction when compared to single-layer reuse and inter-layer reuse, respectively. With larger on-chip memory, the benefit of the hybrid reuse over the inter-layer reuse becomes less because most of the off-chip memory access reduction is achieved by inter-layer data reuse, while the case shown in Figure 6b occupies only a small proportion. With 1 MB on-chip memory, the hybrid reuse requires 0.7% less DRAM access than the inter-layer reuse. It is worth noting that when the capacity of the on-chip memory becomes large enough, the required amount of DRAM accesses would become saturated; however, larger memory requires more power consumption.

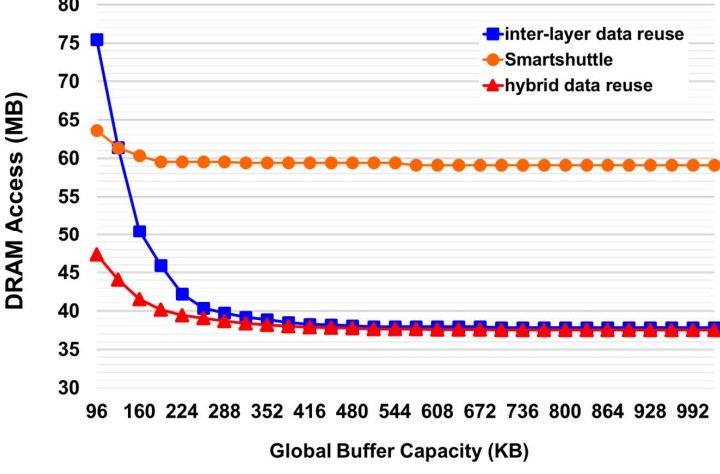

**Figure 7.** DRAM access results of DenseNet-121 with various capacity constraints of on-chip memory.

Comparison results with recent works such as [24,25] are presented in Figure 8 together with SmartShuttle [11]. From the figure, it can be observed that with the same amount of on-chip memory, our method can achieve 46.7% and 51.6% of off-chip memory access reduction when compared to the recent works [24,25], respectively.

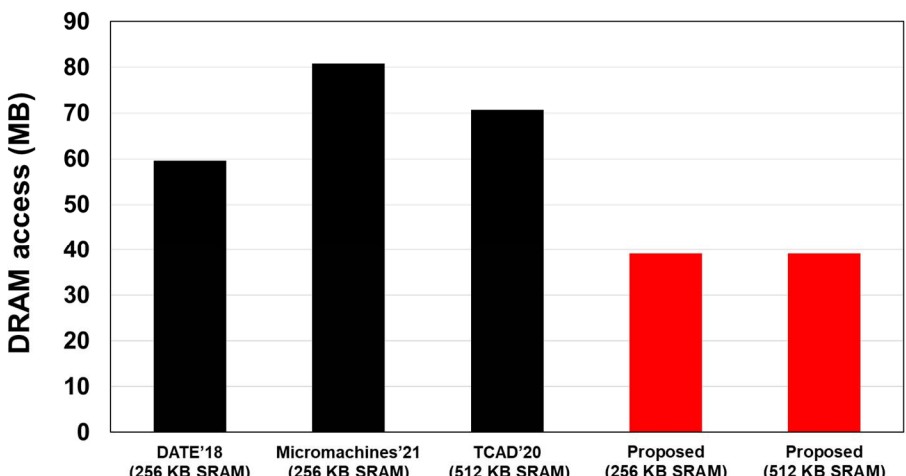

**Figure 8.** DRAM access comparison of DenseNet-121 with previous works.

- **ResNeXt**

The second result is on ResNeXt-50, a DNN model with 32 sublayers ($C = 32$) and 50 layers [4]. ResNeXt is a homogeneous neural network with the Network-in-Neuron shown in Figure 2d. Unlike DenseNet, there are 32 groups in the grouped convolution in ResNeXt.

The result of ResNeXt is shown in Figure 9, which is similar to that of DenseNet shown in Figure 7. For ResNeXt, the intersection point of single-layer reuse with SmartShuttle and inter-layer reuse with two fused layers exists when the capacity of the on-chip memory is 256 KB. When the on-chip memory becomes larger, inter-layer reuse outperforms single-layer reuse, and up to 20.5% of the DRAM access can be reduced. Meanwhile, as with that in DenseNet, the hybrid reuse can always obtain better results than both the single-layer reuse with SmartShuttle and the inter-layer reuse with two fused layers. Although hybrid reuse obtains the same results as inter-layer reuse does when the on-chip memory becomes larger than 480 KB, for the capacity of on-chip memory ranging from 64 KB to 576 KB, the hybrid data reuse approach can achieve up to 20.5% and 66.9% of off-chip memory access reduction when compared to single-layer reuse and inter-layer reuse, respectively.

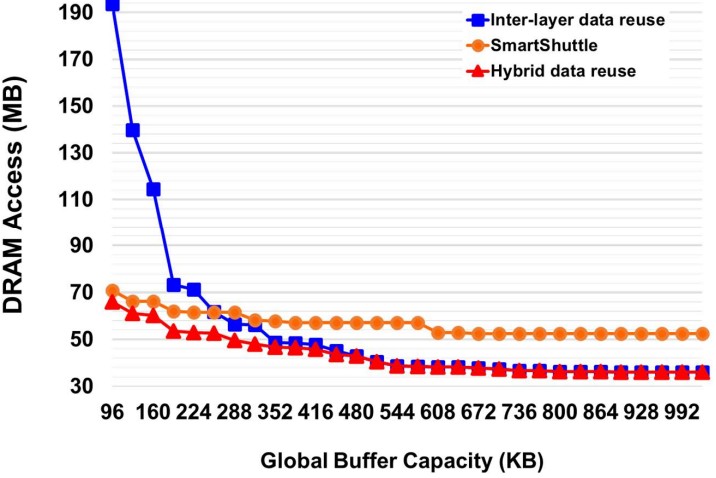

**Figure 9.** DRAM access results of ResNeXt-50 with various capacity constraints of on-chip memory.

- **AlexNet**

The third result is on AlexNet [1], a well-known early neural network that won the ImageNet Challenge in 2012. Unlike DenseNet and ResNeXt shown above, the architecture of AlexNet is irregular; for example, the first CONV layer has a large filter size ($11 \times 11$) with a stride of 4, and the $3 \times 3$ maximum pooling layers with a stride of 2 are added after the 1st, 2nd, and 5th CONV layers, which makes it difficult for optimal dataflow exploration. Therefore, to evaluate the applicability of the proposed method to diverse neural networks, comparison results with SmartShuttle [11] and SuperSlash [19] are given in Figure 10.

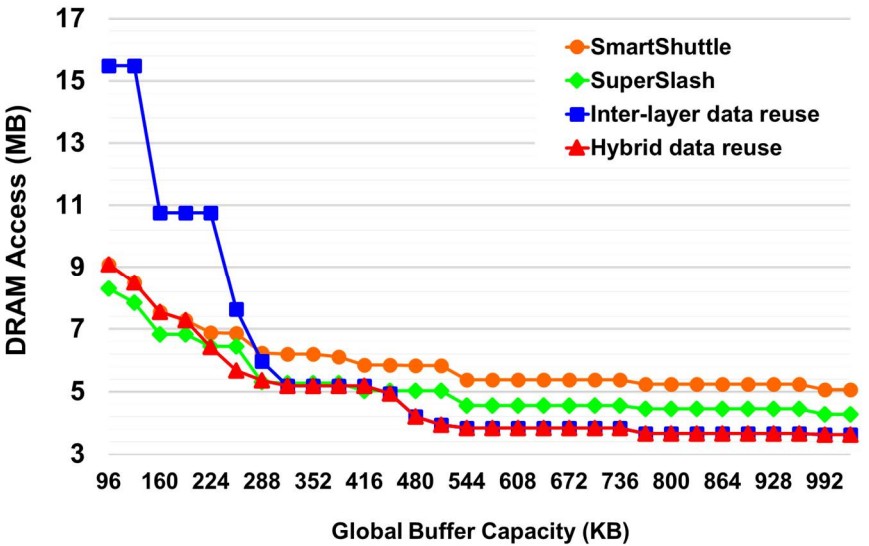

**Figure 10.** DRAM access results of AlexNet with various capacity constraints of on-chip memory.

In Figure 10, SmartShuttle [11] is used as a baseline for comparison, and all the other methods are normalized to it. SuperSlash [19] can obtain better results than SmartShuttle in all the cases, while the proposed methods show significant improvements (up to 17.6%) over SuperSlash when the global buffer capacity is greater than 480 KB. It should be mentioned that with a small global buffer such as one in which the size is less than 224 KB, SuperSlash performs better than the proposed method with 0.2–9.5% of DRAM access reduction. This is because the proposed inter-data reuse method needs to store all the channels of the ifmaps of Layer 1 and one full channel of the filters of Layer 2, which results in it having a smaller exploration space than SuperSlash does. Figure 11 shows the corresponding comparison results for AlexNet with batch size = 1. With 288 KB SRAM, our method can achieve 31.1% and 23.1% of off-chip memory access reduction when compared to the recent works [22,23].

The results of the three modern networks show the effectiveness of the proposed hybrid data reuse approach in minimizing off-chip memory access, where hybrid reuse outperforms single-layer reuse and inter-layer reuse in accelerators with larger and smaller on-chip memory, respectively. The existing edge devices supporting TinyML typically contain less than 1 MB SRAM [6], and the on-chip memory capacity of modern DNN accelerators also ranges from 100 KB to 1 MB. Furthermore, it has been indicated by Han et al. [8] that a common MCU usually has an SRAM smaller than 512 KB (for example, Cortex M7 STM32H743 (512 KB), STM32F746 (320 KB), and STM32F412 (256 KB)). Therefore, it is expected that these memory-constrained devices will benefit from the proposed hybrid reuse approach for more energy efficient DNN processing.

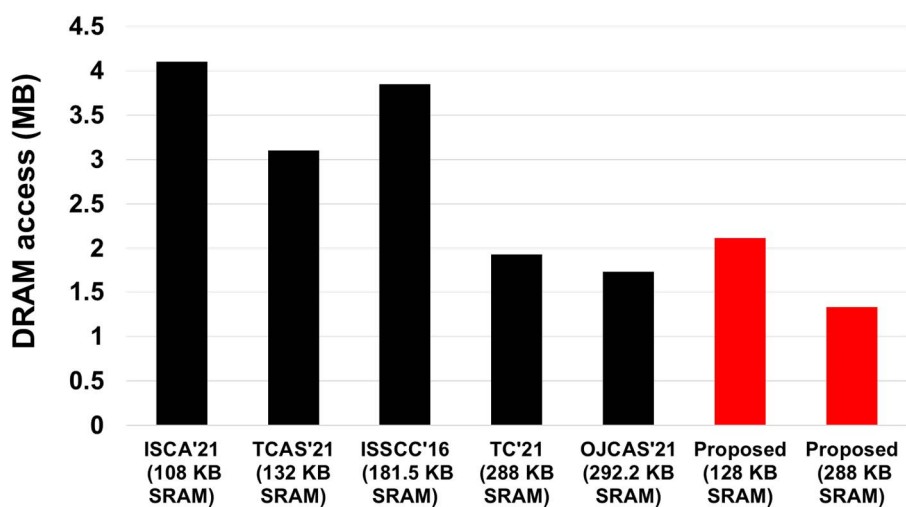

**Figure 11.** DRAM access comparison of AlexNet with previous works (batch size = 1).

## 6. Conclusions

A dataflow optimization method through exploring inter-layer data reuse and single-layer-based data reuse is proposed for modern DNN models. The mathematical analysis of the three inter-layer reuse strategies can be used to precisely estimate the required amount of DRAM access in memory-constrained accelerators. The optimal hybrid data reuse can be determined through exploring the possible single-layer and inter-layer data reuse approaches for off-chip memory access minimization. The evaluation results show that when compared to the existing single-layer data reuse exploration method, SmartShuttle, the proposed hybrid data reuse method can achieve up to 32.5% and 20.5% of DRAM access reduction on DenseNet-121 and ResNeXt-50, respectively, with the capacity of the on-chip memory ranging from 64 KB to 576 KB.

**Author Contributions:** Conceptualization, J.Y. and Y.S.; methodology, J.Y. and Y.S.; software, J.Y.; validation, J.Y., M.Y. and Y.S.; data curation, J.Y.; writing—original draft preparation, J.Y.; writing—review and editing, J.Y., M.Y. and Y.S. All authors have read and agreed to the published version of the manuscript.

**Funding:** This research was supported in part by the KIOXIA Corporation and the Waseda University Grant for Special Research Projects (Project number: 2021C-147).

**Data Availability Statement:** All the necessary data are included in the article.

**Conflicts of Interest:** The authors declare no conflict of interest.

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
