# Peer review of "Dataflow Optimization through Exploring Single-Layer and Inter-Layer Data Reuse in Memory-Constrained Accelerators"

_electronics, doi:10.3390/electronics11152356_

Round 1
Reviewer 1 Report
In this paper, for modern DNN models, a data flow optimization method is proposed by exploring inter-layer data reuse and single- layer-based data reuse, The optimal hybrid data reuse is determined by exploring possible single- and inter-layer data reuse methods that minimize off-chip memory accesses. Compared with existing data reuse exploration methods, the proposed method has a certain improvement. This is an interesting research paper. There are some suggestions for revision.
1. The motivation is not clear. Please specify the importance of the proposed solution.
2. Please highlight the contributions/innovations in introduction.
3. Most of references are a little bit out of date. Please discuss more recently published solutions, especially the solutions published in 2022 and 2021.
4. How much does an off-chip memory access take compared to the corresponding multiply-and-accumulate computation? How much does off-chip memory access typically account for the total network inference time? It is recommended to present quantitative analysis results in the article.
5. What is the relationship between the amount of off-chip memory access among layers and the adoption of an architecture with unified buffers?
6. What is the premise for two or more consecutive layers to be fused together? Can any two layers be fused together?
7. The time complexity of the proposed solution should be discussed.
8. Please specify how to select the suitable parameter values.
9. What is a grouped convolutional layer and how does it translate to other ones such as the normal CONV layers in VGG and the Bottleneck block in ResNet?
10. It is recommended that authors add a lightweight network as an experimental object in the comparative experiment, such as MobileNet, and the method in Reference 20 can be used as a comparative method.
11. Why is SuperSlash not used as a comparison experiment object in the comparison experiment of ResNeXt?
12. The experimental results are not convincing. Please compare the proposed solution with more recently published solutions.
Reviewer 2 Report
Dear Editor,
The manuscript entitled "Dataflow Optimization through Exploring Single-Layer and In-2 ter-Layer Data Reuse in Memory-Constrained Accelerators" by Jinghao Ye et al can be accepted in the present form. The manuscript deals on dataflow optimization for the neural networks. The study is explored from single-layer and inter-layer data reuse approaches
The manuscript is written well and can be considered for publication
Author Response
Thank you very much!
Reviewer 3 Report
The topic to be discussed in this paper contains the intention to solve the memory problem of data recycling.
Abstract of the manuscript to support this may not be summarized well in its entirety. In particular, the original intention of the experiment and the intention to conduct the experiment part is not properly addressed. Since it is not included, specific contents must be also included in the Abstract part.
Also, it is good to use DNN to illustrated for each layer and methodology.
However, the methodology part of DenseNet-121, ResNeXt, and AlexNet to be compared in the verification step in Chapter 5 is understandable.
Since there is no table to express/summarize this, the comparison of the proposed methods with each of DenseNet-121, ResNeXt, and AlexNet - maybe included.
To this end, the author maybe clarify the comparison results including his/her proposed method with other AI network.
Reviewer 4 Report
A dataflow optimization method based on inter-layer data reuse and single layer-based data reuse is proposed for contemporary models of deep neural networks. The mathematical analysis of the three inter-layer reuse approaches is used to evaluate the required amount of DRAM access in memory-constrained accelerators.
The paper can be accepted in present form.
Author Response
Thank you very much!
Reviewer 5 Report
The author proposed a hybrid data flow optimisation method for the modern DNN model. The article is well-structured and rational. The analysis of the optimal hybrid data reuse has been determined and the reduction of DRAM access of 32.5% and 20.5% has been achieved. Therefore, the article is enough to satisfy the journal's readers after a simple minor process as follows:
1) The reduction of DRAM access has achieved the range from 64KB to 576KB with the hybrid data reuse. And so the importance of the range 64KB to 576KB should be described in the article.
Round 2
Reviewer 1 Report
There is a lot of improvement since last version. The time complexity of the algorithms shown in Fig. 4 should be discussed. More solutions published in 2022 should be discussed.
